# Promoted Urbanization of the Countryside: The Case of Santiago's Periphery, Chile (1980–2017)

**Víctor Jiménez Barrado \***[ID]**, Javiera Larraín Suckel, Bárbara Trincado Olhabé and Francisco Cabrera Cona**

Instituto de Geografía, Pontificia Universidad Católica de Chile, Santiago 7810000, Chile; jzlarrain@uc.cl (J.L.S.); bctrincado@uc.cl (B.T.O.); fjcabrera@uc.cl (F.C.C.)

\* Correspondence: victor.jimenez@uc.cl; Tel.: +56-2-2354-4755

**Abstract:** Urbanization of the countryside affects rural areas, especially in the immediate surroundings of large cities. Normally, this occurs as an unpromoted process, but in Chile, it is driven by the legal framework. This research focuses on rural residential plots (RRPs) around the capital city, Santiago. The analysis seeks to understand the significance and consequences of RRPs during the last four decades and the role of a favorable legal framework in affecting their development.'By examining data and official cartography on rural residential plots, the analysis shows a large phenomenon of rapid RRP development in the Metropolitan Region of Santiago de Chile (MR). The study confirms the existence of an ongoing process that is still partially latent and potentially both uncontrolled and evolving. This work demonstrates the negative effect that land liberalization policies can have by promoting territorial transformations that policymakers cannot subsequently control. The conclusions provide a critical perspective on the counter-urbanization process in the context of fragility and scarce resources.

**Keywords:** counter-urbanization; countryside urbanization; neoliberalism; periurban spaces; residential plots; rurbanization

---

## 1. Introduction

In theoretical terms, urbanization of the countryside involves rurban spaces [1,2] or periurban areas [3]. It does not correspond to a concept, but rather to a set of words that refer to a material fact and a practical issue. There is no clear separation between the periurban and rurban concepts [4], so the urbanization of the countryside is an intersectional phenomenon even if the building density and the distance from the city are considered for reference. Its origin is linked to counter-urbanization processes, that is, it is related to the comparison of urban and rural attractiveness when choosing a place to live. This generates a redistribution of the population from urban to rural areas [5]. Depending on its particularities, urbanization of the countryside can be considered a pro-rural process, but also as the opposite trend [6–9]. Without doubt, countryside urbanization has opened an intense debate centered on its consequences. There are clearly positive impacts, such as local economic development, job creation, and increased government tax revenues [10]. This process also generates various negative consequences depending on the geographic conditions in which it happens. Therefore, the spatial context is the key to effectively valuing it.

The context of a Mediterranean climate is very fragile in particular [11]. Urbanization can have a critical impact in areas with this type of climate. This is due to the problems that urbanization causes, like excessive spending, pollution, waterproofing of soils [12,13], increased runoff, and associated risks [14]. An example of the effects of urbanization that are most strongly felt in areas with a Mediterranean climate is the expansion of heat islands caused by periurban spaces [15,16].

Another example is the irreversible loss of fertile and productive land caused by urban growth, especially for agricultural soils [17]. Together they contribute to a mismatch between population growth and land consumption, especially in areas of dispersed urbanization [18]. The consequences of these trends in Mediterranean regions lead to irreversible phenomena such as desertification [19] and the loss of biodiversity [20]. In addition, this urban model demands more resources than an alternative compact model, so it is less advisable when there is a shortage of resources. Notably, land itself is also a finite resource [21,22].

One of the greatest problems of urbanization is that urban use renders the land useless for other purposes. This has been seen in the central zone of Chile, where urbanization of the countryside has taken place in the area with the highest proportion of fertile land in the country. In fact, even today the Metropolitan Region of Santiago de Chile (MR) continues to be the most useful region for vegetation within the country [23]. However, this area suffers from major environmental constraints such as drought, earthquakes, pollution, etc., along with high demographic pressures caused by urbanization. Consequently, urban development increases pressures on the environment. From this, it follows that urbanization predominately does not occur in accordance with physical conditions and the scarce resources offered by the territories it occurs on.

Hence, when the negative consequences outweigh the positive ones, the phenomenon becomes a problem. However, that does not slow down the urban development process. Although these issues also affect the residential function (or are caused by it), this urban use is enhanced by its higher profitability in a liberalized market [24]. So, understanding land exclusively as an economic resource creates risk. Despite this, it is important to highlight that the market economy has played and still plays a fundamental role as a trigger for urbanization [25]. The MR is a good example of the geographic and economic contexts mentioned above.

As mentioned, urbanization of the countryside represents examples of rurbanization and periurbanization. According to the literature, rurbanization and periurbanization are mainly driven by individual choices [26] that underestimate externalities and personal consequences (for example, suffering a socio-natural disaster). For that reason, personal decisions are not based on territorial resources and spatial analysis. More often, these are conditioned by the social, economic, and cultural environment.

The old conception of the countryside (driven by its production value) has changed for most of society and now there is a new one viewing it as a place to realize a rural dream or experience the rural idyll [27,28]. Due to a misunderstanding of the concept of the rural idyll by urbanites, the rural natural space then becomes the perfect place for urbanites to purchase a second residence or build a home [29]. It follows that counter-urbanization is one of the dominant forces of urban expansion [30–32]. This assertion must be understood in terms of urban dispersion and space transformation since its imprint on population flows is much smaller [33]. In addition, in the case of Santiago's periphery, it is necessary to remember that counter-urbanization is a phase that occurs after urbanization [34,35]. Therefore, its magnitude will depend on the size of the urban center where it originated from. As a result, it is more common to find the counter-urbanization phenomenon associated with more urbanized spaces [36]. The centralized Chilean state reinforces this process in Santiago de Chile because the growth of the capital city is also based on its economic and political power [37].

The new urbanization trends followed by a middle class of urbanites seem to be triggered by individual criteria [38] such as diversification of job opportunities offered by a rural area [39] and access to a better quality of life for families [40]. Therefore, individual and family economic preferences are superimposed over the possible shortcomings of peripheral locations. For example, existing public infrastructures are less relevant when choosing a location to live for urbanites that are looking to experience the rural idyll or are searching for rural job opportunities [41]. Conversely, the traditional conception of a rural idyll for rural people is linked to the usefulness of the land for the primary sector. Thus, three basic profiles of people defined by their motivations arise: aspiring farmers, country life lovers, and recreation seekers [42]. The first option occurs in the context of the traditional concepts of

rural life and the other two occur under a different concept. This produces a clash between two basic conceptions of the same space. This tension is usually resolved in favor of the real estate market for economic reasons.

This justifies the debate on the urbanization of the countryside needing to turn toward the causes of its origin and its reproduction. In addition, its management should be a central part of the discussion.

When urbanization of the countryside has created problems, the role of governments in addressing or reinforcing these problems has varied greatly. One solution has been demolition and eradication, which is more common in slums and squatter settlements or in vulnerable areas [43,44]. In this case, the origin of these spaces is related to marginality. Another alternative has been legal battles between administrations/organizations and urbanization promoters [45,46]. The most common solution to disputes over urbanization in the developed context has been passive acceptance and tolerance [47–49], and finally legalization of dwellings [50,51]. In this case, the middle class plays a fundamental role in the origin of urbanization and contributes to moderating its effects. Last, there are extreme market-driven experiences that directly promote this urbanization process and ignore all of its problems. These kinds of urban processes are legitimized under the neoliberal point of view [52].

In the context of triumphant neoliberal ideas, the actions of the state in facilitating the urbanization of the countryside relieve the tension for urbanization supporters between the conceived space and the lived space [53], that is, between what is wanted and what is planned. However, including the countryside in the real estate market can trigger unpredictable investments by capitalists (which can include anyone living under that model) [54,55]. In the study area (the MR), this was already a market worth USD 400 million per year according to official data [56]. The reason for this magnitude of unpredictable investments is that what people seek is shaped by neoliberal state policies, including territorial ones [57].

In Chile, space is the reflection of state policies promoting radical and vertical individualism [58–60]. In other words, it promotes social segregation based on the economic capacity of each individual. In this context, personal choice prevails over land planning analysis. In other words, economic priorities beat environmental and traditional priorities in the countryside.

These trends have occurred in Chile since the approval of Decree 3516 in 1980 [61], the same year when the Constitution of the Pinochet dictatorship was approved. Both legal bodies still remain and contribute to the consolidation of a neoliberal economy. The application of the neoliberal approach to land management has caused adverse consequences. The land liberalization policy restricted the access of lower economic classes to the cheapest land, creating social segregation [62]. To avoid that, some public lands have been used to allow access to housing, but the result has been increased densification and overcrowding [63]. The irregular and precarious settlements developed as a result of this were displaced but not eradicated [64]. So, there was a reduction in habitat quality and rising housing prices, producing inequality, and other related problems as a result. All of this together reopened the gap between the conceived space and the lived space. This is a cyclical process, in which the tension between supporters and opponents of urbanization relaxes when the countryside is urbanized but increases when those spaces are fully urban.

Currently, Santiago's periphery is an "escape route" in many ways and for different actors. It forms a new residential habitat that allows a confluence between the desire for suitable real estate (for the middle and upper urban classes) and the activities of the construction sector. In this context, no actor considers the consequences of this development on a resource-poor environment. Due to this, Santiago's periphery is a spatial representation of radical individualism (privatization of profits and socialization of losses) [65,66]. This is an individualism whose principal motivation is "the possibility of doing". This means that it is not based on necessity, desperation, or the lack of alternatives [67]. Quite the contrary, it is the maximum expression of social and cultural values that rely on the logic of accumulating material goods and services [68].

Given the fact that the Chilean state promoted neoliberal principles, the present study proposes to cover Decree 3516 as a key part of this model. The decree's current validity seems to be a sign of the

acquiescence by democratic governments to an economic model inherited from a dictatorship. In the words of Moulian [69], through different legal mechanisms implemented by the military government, an "iron cage" was built that has limited and subordinated Chilean democracy to economic parameters and cultural values such as individualism, consumerism, and materialism until it became a type of political determinism. Therefore, any public action against this model would be unexpected and would also be in a position of weakness.

The Chilean case is not unique, but it is rare. The neoliberal state has an essential function as both an instigator and a legitimizer of the status quo [70]. This fact, like the process of urbanization of the countryside itself, has not usually been criticized. This is because, in Central Europe and North America, where the first and most of the subsequent studies about these issues come from, urbanization of the countryside is not counterproductive to the physical environment. Liability is not questioned, because the process does not have serious negative consequences: urbanization does not consume scarce resources or valuable spaces. The discussion there is mainly on the cultural changes that it generates. The novelty of this study is a critical analysis of urbanization of the countryside, taking as a premise that it can bring about a harmful transformation in a geographic context, such as in the central zone of Chile (and others). The scattered spread of urbanization in other Mediterranean regions could have the same consequences. Then, the hypothesis of this study is that the development and continuation over time of the same economic model have promoted a transformation of the territory without considering its capability to assume such a transformation. Also, urbanization of the countryside is increasingly breaking the law and occurs outside of the control of the national government. The illegality of the phenomenon justifies the need to study it. Therefore, it seems that a liberalized land market can have consequences for sustainable land-use planning. In order to assess this, this study aimed to determine the extent and magnitude of this land use in the MR, first by analyzing Decree 3516 and then by examining official cartography (contrasted with sample aerial photographs). The official nature of the information sources is an additional sign of the close relationship between state institutions and the urbanization of the countryside. The use of official data from institutions to discuss the model and government discourse is also a novel point. However, this type of analysis can only be replicated where urbanization of the countryside has reached a degree of normalization, as has occurred in Chile.

## 2. Materials and Methods

### 2.1. Legal and Academic Literature Review of Decree 3516

The urbanization of the countryside on Santiago's periphery was triggered by a legal body. For this reason, the first methodological stage must lead to knowledge of the legislation. The main sources of data were the Official Journal of the Republic of Chile and the legal repository of the Chilean National Congress Library. The latter was particularly useful in providing not only the original version of the approved text, but also its modifications, chained regulations, and status. It also contains reports from the relevant ministerial committees and the entire law parliamentary history, including related debates and specific voting. The analysis focuses particularly on guidelines that have a spatial impact, such as definitions of plot size and use or building possibilities. In this study, sanctions and urban discipline issues are set aside (with the possibility of being addressed in future studies). This decision was made due to the impossibility of having free access to the sanctions register and cadastral cartography, which would be completely necessary to verify compliance with relevant norms.

Each article of the decree was examined to determine the technical details provided in this section and the results section. Currently, the legal decree has been in force for 40 years, so its consequences are visible. This has guided the subsequent search for scientific literature on specific casuistry (e.g., results on the structure of a property and its size, possible changes in land use, etc.). With a similar objective, the communication channels of the government and corresponding ministries were also used to determine various policies related to this subject.

After the legislative search came the academic search. Although urbanization of the countryside is theoretically a broad subject, relevant scientific information relative to Chile can be found quite easily. This is due to the common and specific Chilean phrase "parcela de agrado" (rural residential plot (RRP)). This refers to rural property 5000 m$^2$ in size resulting from Decree 3516, which allows the building of country homes. Furthermore, although such a rural plot may be used as a main residence, its original purpose was for leisure and pleasure. The nomenclature of Decree (3516) also served as keywords in the search.

The academic review, using the same strategy in the main databases (Web of Science and Scopus), was not very productive. The search was expanded by entering more generic terms ("suburbanization", "countryside urbanization", "urban sprawl", "rurbanization") together with toponyms such as "Santiago" and "Chile" and was not successful either. Then, as a preliminary conclusion, it was determined that the Chilean case has not fully entered the international literature on the urbanization of the countryside.

Ultimately, the Spanish-speaking scientific databases continued to be more useful for understanding this process in Chile. A significant amount of information was found as a result. Later, a "snowball" method proved useful here.

The analysis sought full coverage of the scientific production on the urbanization of the countryside in Chile and was not related to the one carried out for the theoretical framework and introduction. From this result, texts with redundant information were discarded. Relevant information was extracted from each text that could not be extracted from official data. Fundamentally, a very selective compilation was made with technical information, despite any academic origins. Examples included data on the occupation of agricultural land and the evaluation of this legal body in agricultural terms. In this way, an assessment of the undeclared side effects of the decree was achieved. This in turn complemented the official data, which cannot be used alone to draw various conclusions.

## 2.2. Compilation and Data Analysis; RRP Official Cartographic Data for the MR

As indicated in the Introduction, Chile is a country that has legally endorsed and actively promoted the urbanization of the countryside. For this reason, and unlike other parts of the world, a long and complex methodological process to find, locate, and quantify this phenomenon is not needed.

The National Institute of Statistics (INE) classifies the RRP as a census and territorial category within rural areas. This started recently, in 2002 (22 years after the approval of Decree 3516). In fact, the main official source for this information is the National Census (2017), specifically its micro database. This information contains a double format for the MR: first, basic cartography to locate these settlements by means of polygons that mark their contours (polygon topology). However, the micro database does not report on the exact location of each dwelling, nor on its typology or year of construction (Figure 1). As a handicap, the two censuses are not comparable due to changes in the technical definition of the RRP (also depending on demographic evolution). Second is alphanumeric data associated with the mapping, indicating the number of persons, dwellings, and households involved. The main limitation at this point is the wide temporal separation of the data and the use of a demographic database for urban purposes. As there is no orthophoto of the same year (2017), we cannot contrast the official data on buildings. In demographic terms, it is impossible to replicate a study with the same characteristics due to the great magnitude of the phenomenon.

The urban dynamics of those spaces can only be indirectly inferred. To accomplish this, a mapped database (topology of points) on building permits in the MR is essential. This type of planning permit depends on municipal Public Works Departments (offices that issue and register such permits). This information (compiled and published by INE) is much more complete than the previous information, although less specific on the study subject. The database contains the positions of buildings (with high precision but varied quality), the county (or commune) to which they belong, the year of application, their surface area and typology, as well as their use. As a result, a selection can be made containing isolated buildings of residential use on rural land or those belonging to the census category

of RRP (the latter by means of GIS analysis combined with the previous cartography). Unfortunately, the existence of illegal processes such as the construction of houses under other types of permits (to build warehouses, garages, barns, etc.) cannot be detected. During the preparation of this work, quarantines prevented exploratory studies to verify some of these issues.

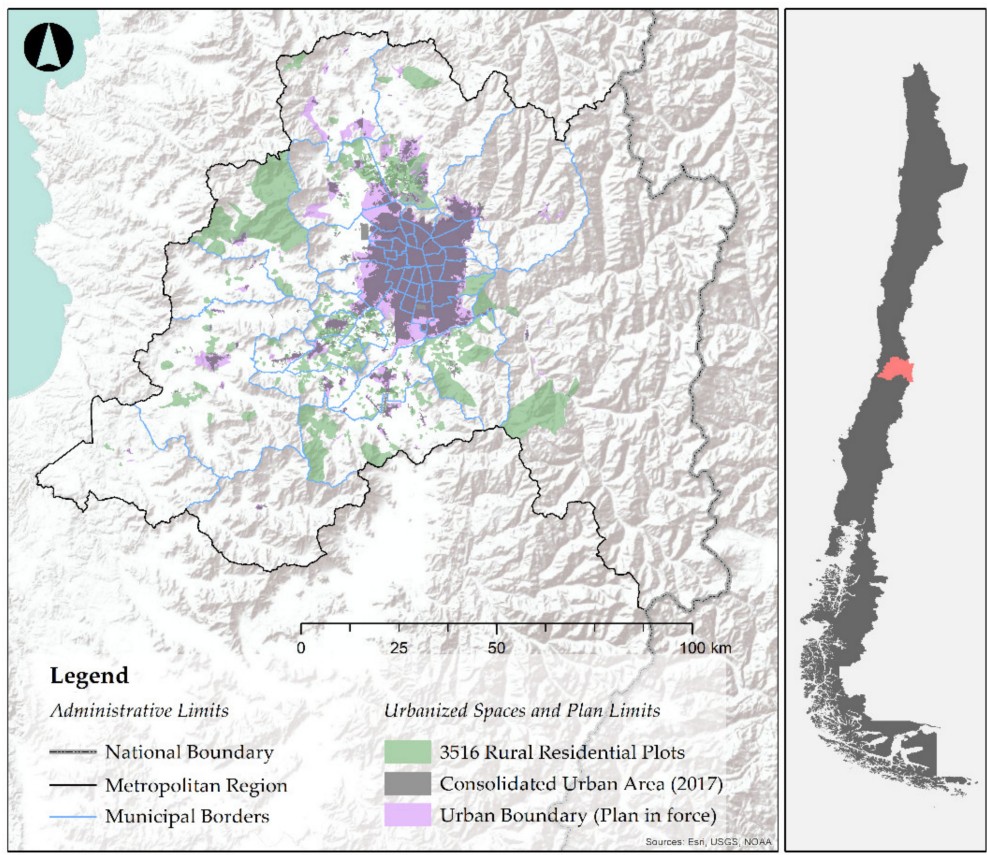

**Figure 1.** Location of rural residential plots (RRPs) in the Metropolitan Region of Santiago de Chile (MR). Source: Own elaboration, based on INE (2017).

Additionally, due to their spatial accuracy, the results can generally be compared with available satellite images (because of the high cost of higher spatial resolution orthophotographs in Chile, ≈0.25 m/pixel, this work uses the most recent satellite images from Google, with compatible spatial resolution).

Therefore, both databases together allow for a more accurate description of the urbanization process affecting the MR countryside, although with limitations. Despite being a phenomenon that began in 1980, its external limits have only been known since 2002. Before that, both vector cartography (cadastral and land planning maps) and aerial photography were very expensive and did not cover the entire study area. It must be added that the criteria used in the census to define them have been changing. In addition, data on building permits are only available from 2010 to 2018 (in the forms already mentioned). Therefore, although present and future analysis can be guaranteed if technical standards are maintained, the retrospective study is based on important constraints.

## 3. Results

### 3.1. Study Area, Legal Framework, and Specific Characteristics of Rural Urbanization in the MR

The Ministry of Agriculture of Chile (Teatinos 40, Santiago) approved, through Decree 3516 in 1980, the possibility that rural plots can be subdivided up to a minimum area of 5000 m$^2$. This standard was applicable to the periphery of the three major cities: Santiago, Valparaíso, and Concepción.

According to the government in charge (Pinochet's dictatorship), the aim behind this legal framework was to avoid the accelerated process of rural exodus, allowing farmers to divide their land among their descendants and maintain agricultural production activity [71]. In the middle of the agrarian counter-reform process, it was argued that the legal process for subdivision of plots involved too many complexities, reducing farmers' access to credit, which in theory could reduce their agricultural productivity [72]. Decree 3516 facilitated this process and in some areas even promoted changes in production patterns to improve the profitability of the soil in smaller plots (the olive crops in the Azapa Valley was an example of success) [73], although these were exceptional experiences.

The most common consequences are quite different. They were unforeseen or at least were undeclared intentions by the government. From the agrarian point of view, the decree incentivized plot atomization, which led to unproductive smallholding [74], despite occurring in an agro-exporting market sector such as Chile. Notably, the minimum plot size was 40 times larger [75] under the previous decree (Supreme Decree 752, 1974). Besides, the government also did not take into account (or ignored) that this process would be accelerated by further hereditary subdivision of plots following the approval of the new decree [76]. In this context, it has played a fundamental role, in that the new generations do not understand the productive value of rural land.

Land-use policy seems to contradict the economic policies established since 1980. This norm hinders the existence of large plots and encourages construction on the best agricultural soils in Chile [77]. Also, the increased value of real estate development (also involving international capital) has encouraged land-use changes on the periphery of the capital city and MR. The latter hosts 40.5% of the country's population (7,112,808 inhabitants) [78], which ensures strong demand. Through this process, land prices have skyrocketed. Agronomic analyses show that the MR has an average property size much smaller than the country's average, but its prices are also much higher [79]. Other econometric studies on the value of agricultural land [80] do not even consider RRPs in their analyses. In fact, according to the Office of Agrarian Studies and Policies (Teatinos 40, Floor 7, Santiago de Chile Mailbox 13.320), these types of plots do not represent the real value of agricultural land [81].

Decree 3516 prohibited land-use changes and, therefore, residential buildings on these plots, but there were legal loopholes allowing construction. These were used over the last 40 years. A practical example is that the construction of housing was possible when necessary to support an agricultural activity (housing for owners or employees). The lack of clear standards to justify this link between housing and plot (size of the plot, farm type and size, housing format, etc.) has led to residential and urban growth to the detriment of agricultural land in the MR. Although this demand did not arise directly from the decree, because it came about before its development, the decree reinforces it and makes it a common way of doing things [82]. According to official figures, the number of non-agricultural properties continues to grow (Figure 2). This has generated a benefit for municipal public coffers, which receive more from taxes [83]. In this way, it was verified that agrarian legislation promotes a reform that, due to its consequences, should have been promoted by the Ministry of Housing and Urban Development (MINVU Alameda 924, Santiago).

As a spatial consequence of the decree, two urban morphologies arose in the rural space since 1980: one formed by plots and individual dwellings, and the other formed by groups of previous plots that add up to between 10 and 500 dwellings, with a closed perimeter [84], popularly known as condominiums [85]. In turn, this generated two opposed models between cities and rurban/periurban spaces: some central urban spaces increasingly became densified and overcrowded due to the verticalization of buildings, and others showed dispersed peripheral spaces for the middle and upper classes, such as RRPs, whose new enclaves are significantly distant from the urban area of the capital city [86]. Both have contributed to Santiago's excessive growth, to the point of representing a large proportion of the MR. The study area extends to 15,403.99 km$^2$, of which 14.81% is represented by residential and urban spaces (RRP + the consolidated urban area).

Another key element of change was the lack of interest in the rural lifestyles of land heirs or buyers and the inability of the state to control the urbanization phenomenon once it started, which led to its

current extent [87]. Today, RRPs are among the principal causes of agricultural land loss in Chile [88], whose current land-use structure is the result of previous agrarian and subsequent counter-agrarian reforms. (Until the second half of the 20th century, the largeholder/smallholder (*latifundio/minifundio*) system persisted in Chile. This meant low levels of productivity, unequal income distribution, high underemployment, and misuse of resources [89]. For this reason, since the 1960s, structural changes began to take place in agriculture as a result of the agrarian reform [90], especially during the governments of Eduardo Frei Montalva and Salvador Allende. With this reform, the structure of land tenure changed drastically, triggering a process of peasant organization, within the context of rights redistribution promoted by the welfare state. Later, what was called the agrarian counter-reform turned out to be the opposite process based on neoliberal capitalist policies that the country experienced during the dictatorship (1973–1989). The implementation of these neoliberal policies had various stages: (1) dogmatic neoliberalism (1973–1982), (2) pragmatic neoliberalism (1982–1989), and (3) the post-dictatorial period [91]). According to the latest official data [92], the loss of agricultural land (between 2013 and 2018) due to plots in the metropolitan region was approximately 8000 hectares. That is, in just five years, counties such as Padre Hurtado (−13%), Talagante (−10%), and San Bernardo (−9%) lost a significant fraction of their agricultural land due to scattered urbanization. A previous study [93] showed that the total agricultural land area affected by urban and developable land until 2006 was 38,976 hectares. The same government report established a direct relationship between the loss of agricultural land due to urbanization and the reduction in agricultural employment. In parallel, an increase in the price of land was detected in those sectors.

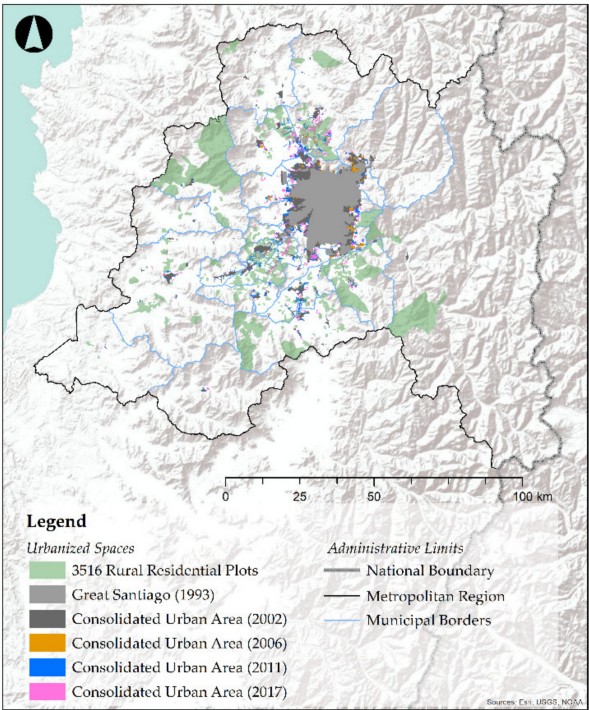

**Figure 2.** Expansion of urban and residential spaces in MR (1980–2017). Source: Own elaboration, based on the Ministry of Housing and Urban Development (MINVU, 2020).

Decree 3516 was definitely a driving force in counteracting agrarian reform [94] by initiating a process of dispossession of the peasants from the land that mainly benefited the agro-exporters and later the real estate companies. This process is still going on. This led, first to a process of proletarianization of the peasantry, and second to an elitization of spaces [95]. In this way, the rurban/periurban areas of Santiago are a clear example of the process of accumulation by dispossession noted by Harvey [96].

Urbanization of the countryside benefits from a laissez-faire policy approach. The lack of land planning in rural areas makes Decree 3516 a de facto land planning instrument [97], although based

on what it really brought about, one could easily define it as a land instrument of "unplanning" [98]. According to Barton and Ramírez [71], there is evidence of a public management strategy under the dictatorship that led to minimizing the role of urban and land planning and ignoring abuses derived from Decree 3516.

Within this framework, since 1980, speculative strategies such as the division of land prior to urban plan approval have been the rule. In other words, there have been ownership structure changes due to the threat that the next urban plan will not allow such changes to occur [99]. In the dictatorship period, and also in the democracy afterward, large real estate companies and also the most important economic and political elites in Chile have been involved [100]. These spaces are located in the "waiting room" for legal urban incorporation, allowing building densification that generates capital gains for their owners [101]. At any time during this period, they are created as RRPs to later be included in the "urban city limit" (a land category where building is possible) and/or to be part of the consolidated urban area (Figure 3). The next step is to "externalize" or simply not cover the urbanization costs involved, that are intended to be imposed on the public [102]. In contrast to this strategy, the state promoted conditional urban development zones (zones that could become urban under certain conditions) from 1997. This allowed for greater building density in rural areas. One of the main conditions for these zones has been that private owners fully assume the cost of roads and equipment for urbanization. In this way, the state intended to combat the attractiveness of RRPs [103]: in exchange for their investment, owners acquire greater security and benefits in legal and personal terms (provides legal security for transmission and inheritance rights). However, this model has not proved beneficial for society as a whole, as it created closed and elitist spaces [104].

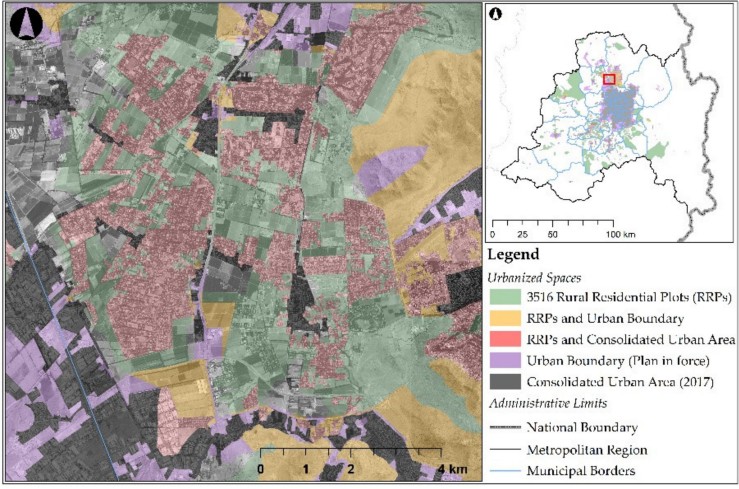

**Figure 3.** The intersection between RRPs and legal city limits in MR. Source: Own elaboration, based on INE (2017) and MINVU (2020).

In any case, Decree 3516 was repealed for the Metropolitan Region in 2006 [105], although this change was not retroactive for plots that had already been divided. Therefore, the previous and intense subdivisions affecting the MR created an extensive "warehouse" of land for the real estate market, which was unevenly occupied [106]. The magnitude is unknown because it is very difficult to access cadastral information. It is only discovered when the phenomenon is evident (houses are already built) and irreversible. The following section presents the most updated figures on this phenomenon affecting the MR.

## 3.2. Extent and Distribution of RRPs in the MR

According to the 2017 census data, 721 sectors in the MR fall under the category of RRP. These sectors are only marked by broad contours, which in turn contain a varying number of 5000 m$^2$ (0.005 km$^2$) plots and dwellings. This type of urban development was found in 20 of the 52 counties of

the MR, excluding those that have a totally urban character. Nevertheless, this list includes counties with a high degree of urban character like La Florida, San Bernardo, and Pudahuel. This is due to the existence of rural or periurban sectors within their municipal limits. In sum, RRPs occupy 1399.95 km$^2$ in the MR, exceeding the number recorded by MINVU for the consolidated urban area (951.92 km$^2$) and that considered within the city limit (863.02 km$^2$).

The number of dwellings in RRPs has reached 28,804, and 85,304 people live there. This means that they contain only 1.20% of the MR population, and they occupy a surface area of 9.09%. This is due to a ratio of people per dwelling of 2.96 and a very low population density (60.93 inhabitants/km$^2$) compared to the regional average (461.75 inhabitants/km$^2$).

Data analysis by county indicates that pericentral counties accumulate a greater number of absolute populations within RRPs: Pirque, Lampa, Talagante, and Colina exceed 8000 residents in these plots, the latter standing out with a record of 18,596 inhabitants. In relative terms, this spatial trend continues to predominate over the total county population. The counties located adjacent to the urban area occupy the first positions of the list (Table 1), although with exceptions like Curacaví (which is more remote and rural). Above the rest are counties such as Pirque (31.46%) and Calera de Tango (26.04%), followed by a large group of counties with a population between 7% and 15% (Curacaví, Talagante, Colina, San José de Maipo, Isla de Maipo, Lampa, and Paine).

**Table 1.** Summary of census information on RRPs. Source: Own elaboration, based on INE (2017).

| Municipality | Inhabitants of RRPs | Housing in RRPs | Area of RRPs (km$^2$) | RRP Sectors | Percentage of Population Residing in an RRP (%) | Percentage of Municipal Area Occupied by RRPs (%) |
|---|---|---|---|---|---|---|
| Pirque | 8343 | 2765 | 154.74 | 69 | 31.46 | 34.80 |
| Calera de Tango | 6613 | 1957 | 27.10 | 74 | 26.04 | 37.01 |
| Curacaví | 4729 | 2465 | 349.49 | 35 | 14.52 | 50.35 |
| Talagante | 9917 | 3188 | 62.33 | 99 | 13.36 | 49.77 |
| Colina | 18596 | 4784 | 149.68 | 86 | 12.72 | 15.42 |
| San José de Maipo | 2129 | 866 | 184.96 | 6 | 11.70 | 3.71 |
| Isla de Maipo | 3104 | 1114 | 26.55 | 46 | 8.57 | 14.00 |
| Lampa | 8638 | 2799 | 62.83 | 56 | 8.47 | 13.98 |
| Paine | 5261 | 2622 | 161.11 | 62 | 7.23 | 23.78 |
| Peñaflor | 5847 | 1774 | 33.89 | 60 | 6.48 | 48.82 |
| Padre Hurtado | 2753 | 863 | 18.54 | 27 | 4.35 | 22.99 |
| Buin | 3854 | 1149 | 33.92 | 37 | 3.99 | 15.63 |
| Melipilla | 3197 | 1387 | 62.87 | 27 | 2.59 | 4.64 |
| El Monte | 774 | 321 | 10.93 | 16 | 2.15 | 9.42 |
| Tiltil | 330 | 285 | 16.82 | 4 | 1.71 | 2.57 |
| María Pinto | 177 | 102 | 8.15 | 5 | 1.30 | 2.07 |
| San Pedro | 117 | 97 | 3.39 | 3 | 1.20 | 0.43 |
| San Bernardo | 535 | 149 | 2.41 | 7 | 0.18 | 1.57 |
| Pudahuel | 273 | 73 | 1.81 | 1 | 0.12 | 0.91 |
| La Florida | 117 | 44 | 28.45 | 1 | 0.03 | 40.04 |

The correspondence between specific population density and surface area of RRPs varies from county to county (Figure 4). There are some with half of their surface area compromised, such as Curacaví (50.35%) and Talagante (49.77%), in which the RRP sectors contain around 15% of their total population, while others with a very similar surface area, such as Peñaflor (48.82%), only have 6.48% of their total population in RRPs. More extreme is the case of La Florida, an eminently urban and small county, showing a wide eastern sector occupied by this type of plot. In this case, the surface area occupied by RRPs is 40.04%, while the population represents only 0.03%. Although the proportion of surface area occupied by RRPs clearly predominates in pericentral counties, the most outstanding value was found in Curacaví, one of the biggest MR counties (leading also in absolute numbers).

When analyzing the ratio of population per dwelling in RRP sectors (a more detailed scale than when using counties), this value is higher in pericentral counties (Figure 5b). Concentrations of sectors with more than three people per house are found in Pirque, Colina, Lampa, Calera de Tango, Talagante, Buin, Padre Hurtado, and Peñaflor. This suggests that those who live there are probably families in their principal home; in other words, the people who live in these counties are permanent residents.

Moving away from the capital city, the ratio decreases and homes with two people or fewer become more common.

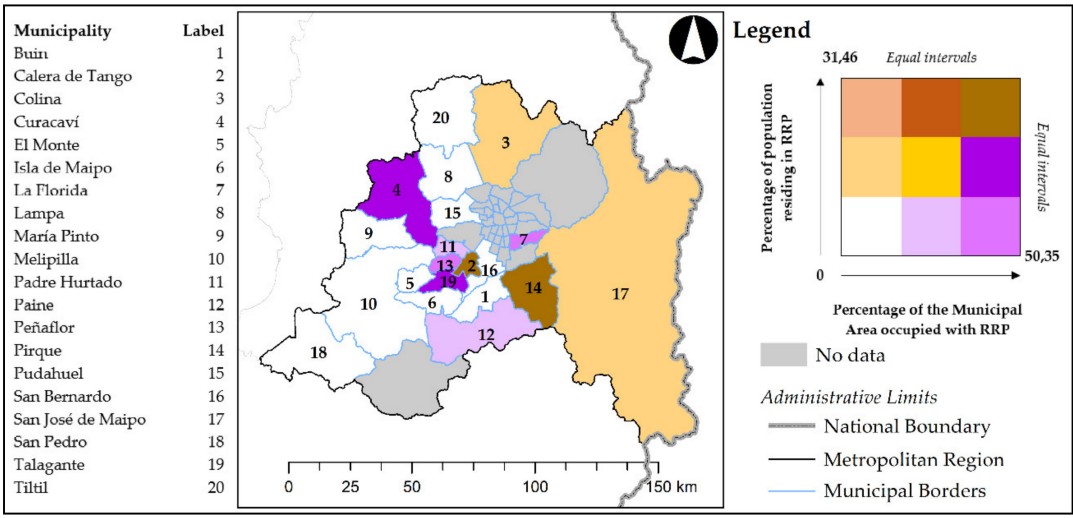

**Figure 4.** Percentage of municipal (county) surface occupied by RRP and municipal (county) population living in RRPs. Source: Own elaboration, based on INE (2017) and MINVU (2020).

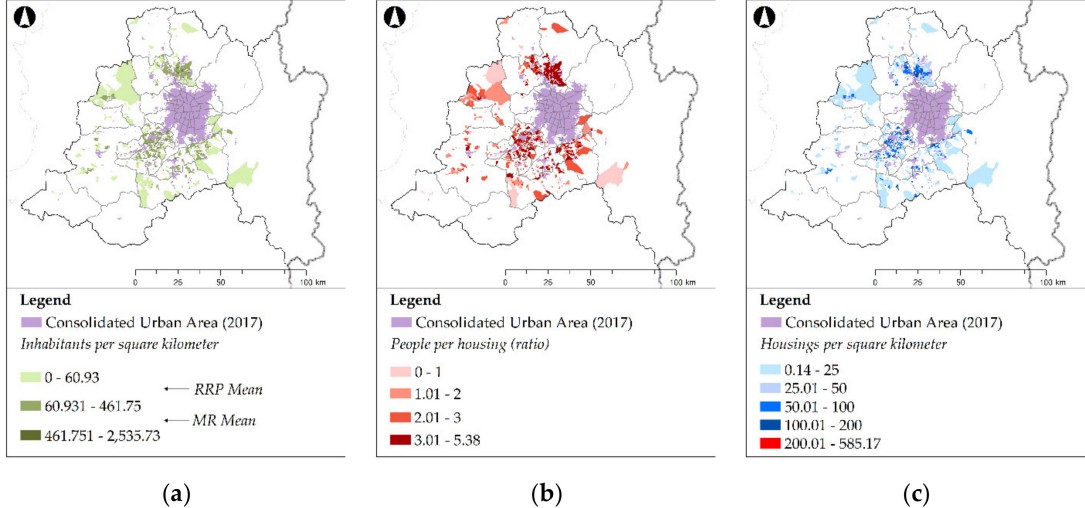

**Figure 5.** (**a**) Number of inhabitants per square kilometer, (**b**) people per housing unit, and (**c**) number of housing units per square kilometer. Source: Own elaboration, based on INE (2017) and MINVU (2020).

Comparing these data with the population density of each sector (Figure 5a), it can be observed that there is a correspondence between these areas. Very few sectors (10.54%) exceed the regional average and are concentrated mainly in the pericentral counties of Colina, Calera de Tango, Talagante, and Peñaflor. If this group is added to the sectors that exceed the average calculated for RRPs, then the resulting group is the same as in the previous paragraph.

However, there are sectors in the first group of counties where a high ratio of people per housing unit does not correspond to the highest population density. This may be due to poor building consolidation of the RRPs and/or an average plot size much bigger than the standard 0.005 km$^2$. The census data show that housing density in these sectors (Figure 5c) is well below the maximum allowed (200 dwellings/km$^2$), so there is no certainty about the reasons for this occurring. Only by using satellite images is it possible to verify the causes: there are sectors with a high concentration of dwellings in certain points, and other sectors without dwellings or physical signs of subdivisions. Therefore, the distribution of built-up areas is very unequal within the space defined as being RRPs.

Also, it can be pointed out that even where the space is not built on, it is very likely that it was legally divided without any visible effects (fences or walls).

Two issues stand out from this information. On the one hand, there is a large amount of land defined as RRP land in steep terrain. This means that there is land where building is technically difficult (even from the legal point of view, because of the socio-natural risks involved). On the other hand, 16 sectors exceed the maximum building density allowed, which indicates a process of illegal densification.

### 3.3. Recent Building Dynamics in RRPs (2010–2018)

The information on building permits clearly describes the most recent RRP urban dynamics. A detailed analysis of housing-use permits shows a changing relative weight of RRPs over the total quantity of land from 2010 to 2018 (Figure 6). The trend seems to indicate that the dynamics of these sectors are unrelated to the general trend, possibly because demand is socially constrained. For the entire period, of the 18,095 permits granted for housing purposes, 1858 were in those sectors (10.27%). This figure is remarkable considering the differences between urban and periurban spaces in terms of their possible building density.

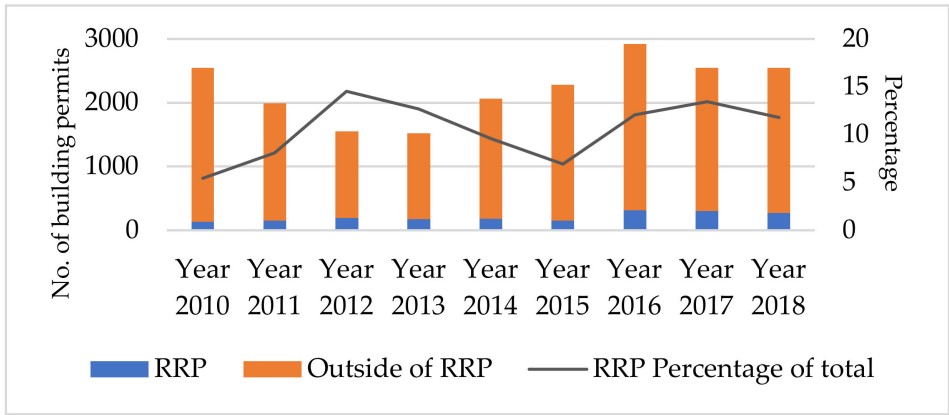

**Figure 6.** Percentage of RRP residential building permits out of the total (2010–2018). Source: Own elaboration, based on INE (2017).

In general terms, the historical data series indicates that there is a predominance of single-story dwellings (56.03%), followed by two-story homes (43.33%), with a negligible presence of residences with three or more floors (0.65%). While the first group displays a very homogeneous spatial distribution, the second group shows a tendency toward concentration in the northern sector of the urban main center, particularly in Colina. The average surface area is 315.57 m$^2$ (219.37 m$^2$ per floor). This means that RRP housing sizes are much larger than the average size of houses in urban counties, even when compared to higher-income sectors [107].

Within rurban/periurban spaces, the spatial distribution based on this criterion does not show clear differences between sectors. According to INE information, the predominant typology is the isolated house. This is confirmed by an exploratory analysis of aerial orthophotographs. These also show the presence of complementary features in individual dwellings such as gardens, swimming pools, and garages.

The data by county (Figure 7a) show that pericentral counties continued to be the most dynamic ones from 2010 to 2018. Particularly important are those located at the northern edge (Colina and Lampa) and southern edge (Peñaflor, Talagante, Buin, and Pirque) of the urban area. They possibly benefited from three main issues: proximity, connectivity, and fewer topographical difficulties (hence their former agricultural use). Colina stands out with 979 permits, with more than half (52.69%) of approved permits concentrated in the study period. Peñaflor and Pirque follow far behind, with 203 and 175 permits, respectively.

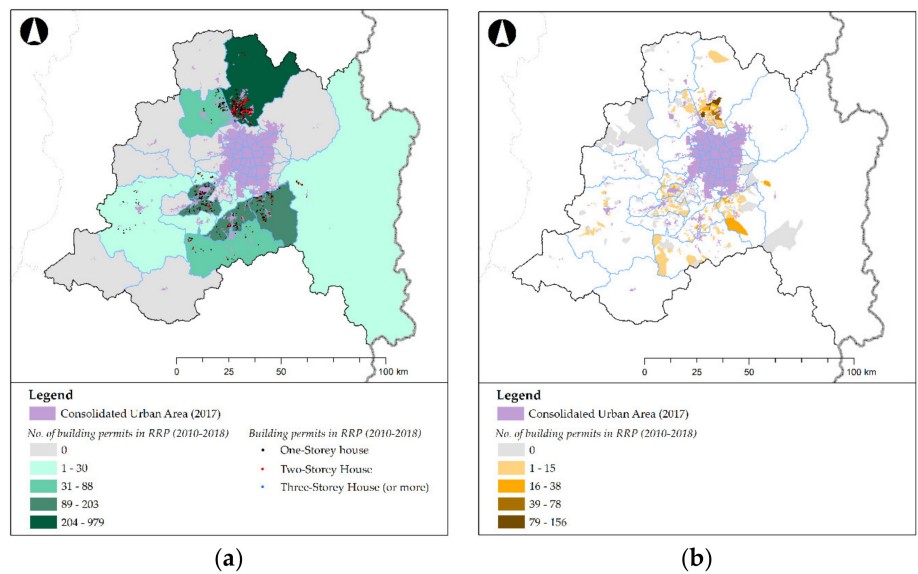

**Figure 7.** Number and location of RRP building permits between 2010 and 2018 (**a**) by county and (**b**) by sector. Source: Own elaboration, based on INE (2017) and MINVU (2020).

The analysis by sector (Figure 7b) shows that dynamics within each county are very heterogeneous. The northern and southern zones continue to predominate, but within these zones, there are sectors in close proximity and even adjacent to one another with opposite dynamics. The orthophotographs show that there are no physical signs or location-related factors that justify these big differences between such close sectors.

In both maps, empty spaces described as "urban blackout sectors" can be observed. Of the 20 counties containing RRPs, 8 have not recorded building permits since 2010 (Tiltil, San Pedro, Pudahuel, María Pinto, La Florida, El Monte, Curacaví, and Calera de Tango). These are mainly western, rural, and remote counties, although there are also examples of such counties near the urban area. Two scenarios are possible: the presence of illegal construction without permits (this cannot be verified, since no satellite images are available dating the construction of these housing units) or the lack or failure of official data. The analysis by sector shows that even in the most dynamic counties there are sectors that lacked permits in the last decade. Aerial orthophotograph analysis indicates three main reasons that could explain this: either these spaces were already consolidated, or they are interstitial spaces not yet developed, or they are areas with physical constraints. Almost two-thirds of RRP sectors (63.66%) fall within this category of lacking recent urban dynamism.

To check the extent of recent urbanization dynamics, the 2018 data on building permits were taken out of the surveyed data. Thus, the records from 2010 to 2017 are comparable to those of the 2017 census. The resulting cartography shows the influence of recent years on the total number of dwellings built on RRPs by sector (Figure 8a). Once again, the pericentral sectors north and south of the urban center take the lead. In other words, the sectors showing the largest number of housing units (Figure 8b) are also those showing the most urbanization dynamics since 2010. This is a clear sign that the process is accelerating and that it is more marked in those places located next to urban centers. At the same time, no growth in residential construction was observed in RRPs located in more rural and remote areas.

Based on the results, different behavior was observed. The urbanization of the countryside crosses local administrative boundaries, so management must have a regional scale. In the most densified sectors, the process is irreversible. For this reason, it is preferable to integrate them legally and fully into urban land. In less densified areas, the solutions must be different. Policymakers should clarify what agricultural use is and the relationship between new housing and agricultural use. This can be demonstrated through records of the activity of the owner and the productivity of the

plot. Territorial and urban policies need to create reserves of fertile soils. In short, this means moving from economic to physical territorial planning. In remote and rural spaces, preparation for urban development is still possible, since many plots in these areas have not been built. This is where the restrictions should be increased: limiting the building typology and the permitted constructed area, monitoring the divisions (especially in plots whose ownership is not related to the primary sector), and increasing vigilance over the urban domain. Additionally, the INE should exclude non-built spaces from its RRP sectors, since they contribute to creating an image of greater availability of land for building.

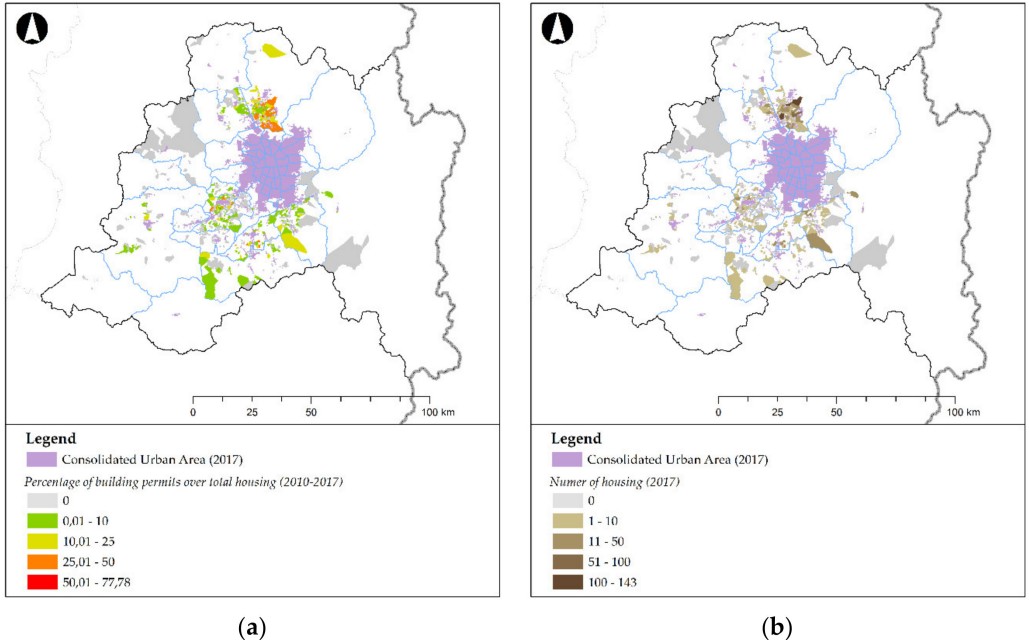

**Figure 8.** (**a**) Percentage of building permits (2010–2017) over total housing and (**b**) total number of housing units (2017). Source: Own elaboration, based on INE (2017) and MINVU (2020).

## 4. Conclusions

The urbanization of the countryside in the MR has been a long-standing process in which the original intention of Decree 3516 (to improve agricultural productivity) has not been fulfilled. On the contrary, the amount and location of housing show that political decisions have been based on ideological/economic reasons rather than technical/agricultural considerations. As in other international cases [108,109], neoliberal land policies have benefited urban advancement and the consumption of territory beyond their capacity. Nowadays, RRPs occupy large areas of former agricultural land and even land not recommended for building (areas of socio-natural risk due to slopes, floods, etc.). Therefore, this study shows evidence that neoliberal legislation puts economic profitability before the territorial capacity to accommodate certain uses.

Furthermore, this case study shows that counter-urbanization is not a homogeneous concept. Urbanization of the countryside can be considered positive when it creates more dynamic and diverse spaces in some contexts, and in other areas can be considered negative when it contributes to spatial competition between land use and the preservation of resources. Consequently, its assessment must always be accompanied not only by morphological and material analysis (of the number and type of dwellings, density, distribution, etc.) but also by consideration of the geographical context where it occurs.

In the study area, there are two dynamics related to the distance to the urban area. Therefore, as elsewhere, there is a gradient in terms of land consumption [110]. On the one hand, there is a dynamic that affects pericentral counties, where spaces have become more dense and have done so

recently. Within these zones, there are some areas with similar physical conditions that have different degrees of urbanization. Again, economic strategies and real estate speculation are key in the selection of sites for investment and building. In any case, areas where the process began before (northern and southern zones) have improved their infrastructure over the years (streets, sidewalks, power lines, connectivity), so it is now more attractive to build on the empty plots that remain. This is one of the possible reasons for the greater impact of the phenomenon there. Undoubtedly, this is a process that progresses by obtaining feedback, but knowing the specific reasons for these changes requires knowing the opinions of owners. This will be a task to be developed in future research.

On the other hand, urban dynamism and densification are much weaker in remote rural areas. There, empty plots predominate along with a small number of old houses in RRPs. While primary residences are common in pericentral areas, rural areas are only sporadically inhabited.

These differences seem to prove that the urbanization of the countryside as a process is explained better by connections with urban centers rather than by connections with rural areas. Therefore, it is a fact that RRPs could be a new way of building a city near Santiago. Its negative outcomes do not make it advisable, so policymakers must control these processes in case they appear in other cities of the world. However, this ability has been a point of separation between the concepts of suburbanization and counter-urbanization. The study shows that urbanization of the countryside can also build an urban environment, but in this case, it is only achieved in very advanced stages of the process. In this way, methodologies based on labeling processes by their current situation could do it incorrectly if they do not consider the evolution of the phenomenon. This study area presents different stages of the same process, which allow us to know what the process is. Without that heterogeneity, in this case study, there would be serious limitations to rolling back the analysis in places with more advanced stages.

To make a more accurate study, in addition to historical analysis, it is necessary to more precisely know the motivations of the owners. Consequently, another methodological process of collecting owner information (currently underway) should be used. The most important limitation is the availability of free, highly accurate satellite images of previous years (for locating and digitizing homes). In the short term, quarantines limit the polling of residents.

In the absence of the above, the available data show that this new way of building the city is not controlled by territorial planning. In fact, illegality plays an important role in the process within the MR. Most worryingly, conditions could get worse. The voracity of the real estate market together with very little market regulation can cause negative consequences in social terms (segregation), as well as environmental problems (occupation of protected natural areas) and personal safety issues (exposure to socio-natural risks). This demonstrates Sen's idea [111] on rational choice theory: individual rational decisions cause collective harm. In addition, it is exacerbated if the political and economic context encourages them instead of limiting and subordinating them to the common good. Lawbreaking is recorded as official information (despite variations in its quality and consistency), therefore, this is the reason why the government has launched campaigns against illegal urbanization. Among them is one named "No more irregular lots", a public reflection of Law No. 21,108, enacted in September 2018.

The prolonged government inaction and its current timid initiative is proof of two things: first, official information needs to be complemented by other methodologies that track illegalities and anticipate the problem. In the neoliberal context, external and independent help will most likely be needed for auditing, including academia. Second, the measures implemented by the government are insufficient, late, and not in line with its real capacity to tackle the problem. Clearly, it supports the view that the free market is the principal instrument regulating space, but this concept creates a broad framework in which urban dynamics can be reproduced and accelerated to such a degree that they go out of public control. This is valid globally. In the Chilean case, this also presents a serious problem of democratic order (it must be remembered that the decree was approved during the dictatorship period) and social space justice, especially when Chile has endorsed various international agreements, such as Goal 11 of the United Nations Development Program, on sustainable cities and communities [112].

Finally, and based on its performance, it is very unlikely that the government will endorse the creation of new methodologies to tackle this issue. In this context, public and academic exposure to the negative consequences of the urbanization of the countryside in the MR could promote a change of course for the government and a new way of managing territory, in which the prevalence of sustainable territorial planning and studies of the environmental impact on the market are respected. Guaranteeing the advancement of urban areas according to the suitability of the territory should be a priority for Chile in the context of current climate change scenarios and the likely greater short-term real estate attractiveness of rural areas due to the pandemic.

**Author Contributions:** Conceptualization, V.J.B.; methodology, V.J.B. and J.L.S.; formal analysis, J.L.S., B.T.O., and F.C.C.; investigation, V.J.B. and J.L.S.; resources, B.T.O. and F.C.C.; data curation, B.T.O. and F.C.C.; writing—original draft preparation, V.J.B.; writing—review and editing, J.L.S., B.T.O., and F.C.C.; visualization, V.J.B., J.L.S., B.T.O., and F.C.C.; supervision, V.J.B.; project administration, V.J.B.; funding acquisition, V.J.B. All authors have read and agreed to the published version of the manuscript.

**Funding:** This research was funded by FONDECYT Project No. 11190058 (Agradourbanización: estar sin ser en el rural chileno. Fiscalización espacio-temporal de un cambio territorial y socioeconómico en la Región Metropolitana), Research and Development Agency (ANID-FONDECYT).

**Conflicts of Interest:** The authors declare no conflict of interest.

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
