# Peer review of "Promoted Urbanization of the Countryside: The Case of Santiago’s Periphery, Chile (1980–2017)"

_land, doi:10.3390/land9100370_

Round 1

Reviewer 1 Report

Thank you for the opportunity to review this paper. I commend the authors for their thoughtful revisions. 

Author Response

----- Reply to reviewer 1 -----

Thank you for the opportunity to review this paper. I commend the authors for their thoughtful revisions. 

Dear reviewer,

Thank you for your statements and the contributions you have made. The suggestions have certainly improved the text.

Finally, we highlight that we have used the MDPI English Editing Service (Specialist Edit - ID: english-22488) to improve the understanding of the text.

Authors

Reviewer 2 Report

The study focuses on the rural residential plots around Santiago, Chile and it seeks to understand its significance and consequences during the last four decades.

The paper is well written and for a better understanding of its logic, some explanations should be added. Please refer to the comments below.

1. Line 43: Pls explain more on Mediterranean climate - urbanization nexus (additional explanations are needed apart of what is already included in line 45);

2. The novelty of this contribution must be highlighted;

3. Also, the contribution of this study to the literature must be emphasized;

4. Taking into account that your research is based on a desk research approach, you should add a substantial paragraph with the limitations of the study; 
5. Based on your results, several practical implications for policymakers must be summarized (so, present them in a synthetic form) at the end of chapter 3.

Author Response

----- Reply to reviewer 2 -----

First of all, we would like to thank the reviewer for his work and comments. Without a doubt, they have made a great contribution to our work.

֍The study focuses on the rural residential plots around Santiago, Chile and it seeks to understand its significance and consequences during the last four decades.
The paper is well written and for a better understanding of its logic, some explanations should be added. Please refer to the comments below.

Thank you very much for your recommendations. They are very pragmatic and synthetically orderly. This has helped a lot when it comes to assuming and carrying them out. We sincerely welcome suggestions for improvement and the way they were made.

We strongly request that you review the Word file and use the “Track Changes” tool (All Markup). However, to check the indicated lines of text, we recommend selecting the "No markup" option.

֍1. Line 43: Pls explain more on Mediterranean climate - urbanization nexus (additional explanations are needed apart of what is already included in line 45);

We have introduced more explanations through examples: we talk synthetically about climatic effects, the relationship between population growth and land consumption, desertification and impacts on biodiversity in Mediterranean regions [lines 47-54]. We believe the connection and the consequences are already clear. On the other hand, we consider unnecessary and even counterproductive to include more text and bibliography.

֍2. The novelty of this contribution must be highlighted;

We have made changes at the end of the Introduction. They appear (highlighted in blue) on line 158, between lines 159 and 168, and between lines 177 and 180. They are related to 3 points: the uniqueness of the Chilean case, the critical approach about counterurbanization process (valid for Mediterranean regions), and the treatment of official data to contradict government arguments.

֍3. Also, the contribution of this study to the literature must be emphasized;

The novelty of the paper is related with its contribution to the academic literature. Because of this, we started by making minor changes to the abstract to warn about the critical perspective of the study [Lines 26-27]. We believe that critical studies on counterurbanization are not predominant, so we have highlighted and explain this aspect (lines 159-168). In addition, in the conclusions we have highlighted that the concept is not homogeneous: its value depends on where the process occurs [lines 545-551]. Also, we point out that through counterurbanization the city is also built. In fact, the most consolidated urbanization of the countryside in the study area is integrated into the city of Santiago. That can clarify some conceptual confusion [lines 568-576]. In the end, we leave a conclusion about neoliberal land management, which can be valid internationally [lines 599-601].

֍4. Taking into account that your research is based on a desk research approach, you should add a substantial paragraph with the limitations of the study; 

In a previous review we removed much of this explanation. We were warned that it was not worth telling limitations and what we had not done. That is why we focus on the methodological process carried out.

To take up your suggestions we have made some changes:

A small new mention is made between lines 178 and 180. This explains a limitation to replicate the study elsewhere, but not our limitation.

But the main contribution is made in the methodological section:

In Chile, access to certain information is not free and research project funds are limited [lines 192-194 and lines 267-269]. In addition, the quarantine has limited the verification of official data [lines 254-255]. Access to cadastral information is very complicated in Chile, that was also remembered on the lines 378-380.

The conclusions also recall some limitations and the following studies and methodological processes to be carried out [lines 579-581].

֍5. Based on your results, several practical implications for policymakers must be summarized (so, present them in a synthetic form) at the end of chapter 3.

An entire paragraph is included at the end of chapter 3 to summarize the suggestions for policymakers [lines 522-534]. To sum up, it is necessary to raise the management scale and carry out differentiated actions. The existence of areas of irreversible urbanization must be assumed. On the other hand, there are other zones where policies can anticipate the consequences described in the text.

Finally, we recognize our language constraints, which has hurt us. For this reason, we have used the MDPI English Editing Service. We choose the “Specialist Edit” (ID: english-22488), which includes all features of the Regular Edit, followed by a check of subject-specific terms and style by an expert in this research field.

Authors

Reviewer 3 Report

The main objective of this article is studying the process of land use in the metropolitan areas of Santiago. In this study the analysis of national policy regulations concerning the land subdivision and use is a central focus.

Nevertheless, although the authors seem to be aware of the limitations of the available data, there are some crucial issues that are simply touched in the study, which deserve more attention and analysis.

First, the process of land use change over the years concerns the most productive agricultural areas and there is no data confirming/describing that the process of urbanisation in rural areas has meant less land available for viable agriculture. The analysis does not provide any evidence of the consumption of the most productive land around the city of Santiago. Could you provide some data on the type of productions/farming existing there? And what did imply for the agricultural sector: the out-migration of family farms and the disappearing of farm production from the market?

Second, some effects on agriculture are unclear and must be clarified: (lines 239-241) "the new land configuration after Decree 3516 has proved successful after changes in production strategies were introduced and for certain crops [66], although these are only exceptional experiences". What does it mean?

Third, other effects of the consumption of agricultural land for other uses are only mentioned, but in a very superficial way: (lines 116-117) "increased densification, overcrowding, reduction in habitat quality, rising housing prices and other related problems". This is too generic and the negative impacts need to be explained much better than this.

Fourth, there is a strong contradiction within the analysis between the following statements:

  • (lines 140-142) "the urbanization of the countryside is increasingly independent from the legality and the control of the national government, in favor of free market dynamics and against sustainable land use planning."
  • (lines 260-261) "Decree 3516 prohibited land use changes and, therefore, residential buildings on these plots, but there were legal loopholes allowing construction".

This means that there is a problem of legality, since if there were effective controls (the problem is not analysed in depth in the study), probably all these residential buildings for urbanisation would have not been put in place in a diffused way. This legality problem arise in most of developed countries.

Finally, the analysis of the process of urbanisation is mostly descriptive, without providing sufficient analysis of main drivers: for example, why are the recent building dynamics concentrated just in the north and south of Santiago? Is there some reasons linked to the quality of land or the infrastructures' networks? 

Finally, the English text needs some revisions and editing.

Author Response

----- Reply to reviewer 3 -----

First of all, we would like to thank the reviewer for his work and comments. Without a doubt, they have made a great contribution to our work.

֍The main objective of this article is studying the process of land use in the metropolitan areas of Santiago. In this study the analysis of national policy regulations concerning the land subdivision and use is a central focus. Nevertheless, although the authors seem to be aware of the limitations of the available data, there are some crucial issues that are simply touched in the study, which deserve more attention and analysis.

We respond in an orderly and individualized way to each of the questions. There are changes beyond the recommendations. We strongly request that you review the Word file and use the “Track Changes” tool (All Markup). However, to check the indicated lines of text, we recommend selecting the "No markup" option.

֍First, the process of land use change over the years concerns the most productive agricultural areas and there is no data confirming/describing that the process of urbanisation in rural areas has meant less land available for viable agriculture. The analysis does not provide any evidence of the consumption of the most productive land around the city of Santiago. Could you provide some data on the type of productions/farming existing there? And what did imply for the agricultural sector: the out-migration of family farms and the disappearing of farm production from the market?

After a better bibliographic search, we have incorporated two official studies. The data speaks precisely of the two issues that are discussed in the suggestion. These reports provide us with numerical data and the consequences of the process on employment and land prices [lines 334-341].

֍Second, some effects on agriculture are unclear and must be clarified: (lines 239-241) "the new land configuration after Decree 3516 has proved successful after changes in production strategies were introduced and for certain crops [66], although these are only exceptional experiences". What does it mean?

We believe that we did not explain it well. We wanted to say that the promised improvements only came in very rare cases. The cited text speaks of the experience of the Azapa Valley, where the Decree promoted an improvement in production patterns and was not used for urban planning purposes. The wording has been modified [lines 283-285].

֍Third, other effects of the consumption of agricultural land for other uses are only mentioned, but in a very superficial way: (lines 116-117) "increased densification, overcrowding, reduction in habitat quality, rising housing prices and other related problems". This is too generic and the negative impacts need to be explained much better than this.

We discuss this in greater detail between lines 47 and 55. New text has been added to address the nexus between urbanization and Mediterranean regions. Also, the wording is more concise in the indicated part [Now in lines 135-137]. We point out the consequences with further analysis and provide new references. The text has been rearranged without making too many additions.

֍Fourth, there is a strong contradiction within the analysis between the following statements:

  • (lines 140-142) "the urbanization of the countryside is increasingly independent from the legality and the control of the national government, in favor of free market dynamics and against sustainable land use planning."
  • (lines 260-261) "Decree 3516 prohibited land use changes and, therefore, residential buildings on these plots, but there were legal loopholes allowing construction".

This means that there is a problem of legality, since if there were effective controls (the problem is not analysed in depth in the study), probably all these residential buildings for urbanisation would have not been put in place in a diffused way. This legality problem arise in most of developed countries.

More than a contradiction there is an error in our explanation. This is probably caused by our translation problems. We have modified the first quoted text. The urbanization of the countryside is not independent from the legality, because, in fact it is breaking the law [lines 170-172].

There really is a legality issue that has not been addressed yet. This is the first article of a higher research project. The text indicates that other methodological processes will be developed to solve this in the short-term future. The data from this research shows that this problem exists (this already means an achievement) and will be addressed in depth in future research.

As for the second sentence, it shows that the decree established limits that due to the picaresque have not been respected. Houses not related to agricultural activity posed as houses related to agricultural activity. Legally the change did not exist, but it really did [lines 304-305].

֍Finally, the analysis of the process of urbanisation is mostly descriptive, without providing sufficient analysis of main drivers: for example, why are the recent building dynamics concentrated just in the north and south of Santiago? Is there some reasons linked to the quality of land or the infrastructures' networks?

Saying a reason with the current data would be unwise. In the text we have ventured that it could be due to speculation and land reservation to improve future profitability. Infrastructure networks are better because they are denser spaces and have been built before. The process feeds itself.

As there is no clear evidence, we prefer to explain the reasons in future studies with new methodologies (currently under development). In any case, a summary of the above has been entered as assumptions [lines 557-562].

֍Finally, the English text needs some revisions and editing.

Improvements have been made. We recognize our limitations in this regard, which has hurt us. For this reason, we have used the MDPI English Editing Service. We choose the “Specialist Edit” (ID: english-22488), which includes all features of the Regular Edit, followed by a check of subject-specific terms and style by an expert in this research field.

Authors

Round 2

Reviewer 2 Report

The authors responded to all my comments. My recommendation is "accept in present form".

Author Response

----- Reply to reviewer 2 -----

The authors responded to all my comments. My recommendation is "accept in present form". 

Dear reviewer,

Thank you for your statements and the contributions you have made. The suggestions have certainly improved the text.

Authors

Reviewer 3 Report

Authors have answered to my comments and I can feel myself satisfied with their revisions.

Only two last comments on new sentences they have introduced in the second version:

  • (lines 567-568): "Therefore, RRPs could be a new way of building a city near Santiago, as well as in other cities of the world". This sentence seems to contradict the results about the spatial policy that is behind RRPs and the free market logic that produced so many negative outcomes. I would be more cautious in saying that RRPs, as they were conceived, can be deemed a new way of buiding a city and proposing this recipe for the rest of the world.
  • (lines 598-601): "Clearly, it supports the view that the free market is the principal instrument regulating space, but this concept creates a broad
    framework in which urban dynamics can be reproduced and accelerated to such a degree that they go out of it." I would say more clearly that "to a such degree that they go out of public control".

Author Response

----- Reply to reviewer 3 -----

Authors have answered to my comments and I can feel myself satisfied with their revisions.

Dear reviewer,

Thank you for your statements and the contributions you have made. The suggestions have certainly improved the text.

Only two last comments on new sentences they have introduced in the second version:

The most current changes are highlighted in grey.

(lines 567-568): "Therefore, RRPs could be a new way of building a city near Santiago, as well as in other cities of the world". This sentence seems to contradict the results about the spatial policy that is behind RRPs and the free market logic that produced so many negative outcomes. I would be more cautious in saying that RRPs, as they were conceived, can be deemed a new way of buiding a city and proposing this recipe for the rest of the world.

We have not been able to explain it well. We want to say that it is a fact. It is the reality although it is not recommended. We have made changes to clarify this [lines 567-569]:

“Therefore, it is a fact that RRPs could be a new way of building a city near Santiago. Its negative outcomes do not make it advisable, so policymakers must control this process in case they appear in other cities of the world”.

(lines 598-601): "Clearly, it supports the view that the free market is the principal instrument regulating space, but this concept creates a broad framework in which urban dynamics can be reproduced and accelerated to such a degree that they go out of it." I would say more clearly that "to a such degree that they go out of public control". 

Thanks for this accurate correction. It is done [lines 601-602].

Authors

This manuscript is a resubmission of an earlier submission. The following is a list of the peer review reports and author responses from that submission.

Round 1

Reviewer 1 Report

Dear author(s),

I enjoyed reading the paper. It presents a substantial empirical material I notice has been carefully gathered during a reasonable period of time. However, there are countless inaccuracies and flaws that include contents, ideas’ progression, but also English, so, I suggest to reshape the entire draft and please: make the English proof reading as you manage the language but there are clear issues of collocation and relationality that do not allow following ideas, conclusions, etc. which finally undermines the whole professionalism of the writing exercise and scientific soundness.   

For example, at the beginning of the Introduction you indicate that: “Normally, this occurs as a voluntary process and not as a planned strategy. However, this work considers countryside residential development as just another neoliberal real estate policy in Chile”. In this phrase the use of ‘however’ does not make any sense. “However” is a connector used to contrast two or more ideas, i.e. used when you are saying something that is different from – or contrasts with – a previous statement. But in your statement, you are not contrasting anything. How can you contrast the supposed causes of “countryside urbanisation” (which you say occurs as a voluntary process…etc.) with the purpose or tenet adopted for the paper? (the fact that you assume that countryside development is the outcome of neoliberal policies). This is the same as saying “trees are green because of the sun, however I’d like to play tennis”. See?

A similar situation is at the line 47, when you indicate that countryside urbanisation is entropic, triggered by transportation, etc…(which seems to be a statement about the characteristics and factors that influences processes of ‘countryside urbanisation’), but then you say ”however” there are different zones defined by different socio-economic groups. No sense! the same applies for the use of ‘although’.

What is ‘countryside urbanisation’? sometimes you use the term ‘urban sprawl’ (different), ‘rururbanisation’ (also different). It seems that ‘countryside urbanisation’ is a created or proposed term (as your do not use the definitive article that precedes the composed term, otherwise it must read as ‘the countryside urbanisation’ or better said ‘the urbanisation of the countryside’). If this is a composed term, so, you must define it and clearly indicate (to justify its use) the differences with urban sprawl, suburbanisation, rururbanisation or any other. Otherwise, why not simply using ‘suburbanisation’ for instance? (or any other?)  

There are also some assumptions that are wrongly presented. You start your Introduction with the phrase “Countryside urbanization has opened an intense debate centering on its consequences. In general, the analysis focuses on the negative impacts such as land use changes, elimination of farmland and space transformation (especially agricultural land)”. Why do you suppose (or assume) that ‘land-use changes’ are a negative impact per se? If done properly, land-use changes can be something good, isn’t it? Or why ‘space transformation’ (whatever that means) is also negative?

Words collocations: big cities or large cities? What ‘the current era high mobility rate’ means? (line 61). What ‘as this concept requires’ means? You maybe wanted to say ‘as this concept suggests’? (line 55). What ‘the dream home’ means? Why ‘densification’ per se is a problem? In line 117 you indicate that a legally binding scenario has restrained state powers regarding controls over countryside urbanisation. It reads that ‘countryside urbanisation’ has a negative connotation but you haven’t explained this before. It seems that this term is pre-defined or carries a negative connotation that you have to clarify better before using it as well-known in the literature. Indeed, ‘urbanisation of the countryside’ (as I guess it could be) is closer to the Sievert’s idea of the Zwischenstadt (in-between) to describe urban sprawl as a rich amalgamation of urban, suburban and rural spaces that creates hybrid zones of interaction in which new forms of urbanisation can flourish. The author – as well as many others – see this as potential for restructuring the peri-urban space of city regions rather than keeping criticising urban sprawl (it is enough of that, isn’t it? i.e. the literature and the evidence is clear about it since the 60s!), so, you can maybe clarify this proposed term better and use it consistently.  

There are tons of statements like these that are not possible to follow, up to your conclusions. What is more, in your conclusions (line 441) you indicate that ‘the original intention of Decree 3516 to improve agricultural productivity has not been fulfilled’, which is not true. The original intention of this legal directive was not ‘to improve agricultural productivity’ (who said that!) as it was always understood that half-hectare land won’t constitute a land for agricultural exploitation due to its size and supply of residential infrastructure. Paradoxically, you indicate something different in your methodology while describing this regulation (line 136-136): ‘…although this rural plot may be used as a main residential place, its original purpose was leisure and pleasure’. So?

The term 'neoliberal' is used in a very superficial way. 

In your methodology, you mention that one limitation in finding information about this very specific Chilean law (3516) related to a searching exercise within the Anglo-Saxon (dominated) databases, in which there are not substantial results (something you can actually expect beforehand). You extensively explain this while instead, give little space to what was better methodologically speaking. I suggest – in terms of rationality – to introduce the methodology focusing on what was useful and productive to gather the information (methods, or what to do) rather than telling a story of what you haven’t done (or what not to do) to gather and analyse the information. It doesn't actually make sense. Because of that, indicating what we don’t do in the methodology is not really common. The most critical point here, is the rational approach on searching data on something very specific (a Chilean law) - reclaiming that there is no information - rather than conceptualising it within a consistent theme that clearly state what’s the research topic (rather than related and superficially addressed ideas around urban sprawl, suburbanisation, etc…not really developed in the paper. You can actually add more terms, like peri-urbanisation, post-suburbanisation, urban-rural transects, urban-rural interface, and others…so, why not?). Instead, you choose another apparently very local term (“rural residential plots") to become the main spine of the paper, while the terms is also not well described in conceptual terms. 

There is no literature review section. Why? This explains why international literature is absent and thus, the wide conceptualisation (and the theoretical contribution) of the research topic and the discussed planning directive 3516 as part of extant literature. If we would like to indicate what is the contribution to the extant literature of this paper beyond the Chilean case, what would be the answer? Can we, for instance, indicate that this paper contributes to wider debates around ‘suburbanisation’? or maybe ‘urban sprawl?, or ‘Peri-urbanisation’? ‘Post-suburbanisation’? all of them? Why? How? Did you previously discussed these terms to contextualise the Chilean case, the RRR and the 3516 law here?

The paper is strongly ‘Chilean’ (case study/empirically based) and as explained, it presents countless flaws. Thus, I cannot support its publication in a more internationally focused journal (as Land is). However, and considering its rich empirical material, I suggest writing it again in Spanish, structure this again and submit the draft to a Spanish-speaker oriented journal.

Regards,

Reviewer 2 Report

This is a thoughtful article examining important phenomena and the authors are to be congratulated for their work. It is well written and thoughtfully designed. The analysis of decree 3516 compelling demonstrates how regulatory governance and government failures shape land uses, with all of its attendant consequences.

This article would be strengthened by further elaboration on the theoretical framework (it seems historical institutionalist in orientation but this is not stated anywhere) and methods: the theoretical framework is not articulated and the methods are missing a description on how the documents were analysed (just a few sentences are needed). The article would benefit from further elaboration of how/why governmental interests are maintaining the status quo. In the conclusion the lack of inaction is in part explained by a lack of data/understanding the scope of the problem. But are there other reasons related to public finance or political appeasement? A bit more elaboration here would be useful. This part of the analysis seems underdeveloped.

Please find a few more additional comments below.

  • Line 45: “newly cities”? (word missing?)
  • 58: perhaps this point is better reframed as a contestation of rurality as opposed to transformation: rural idyll versus productive working space. Presumably the productive value is still in play.
  • In section 2.1 it is recommended to describe not just how the relevant databases were identified and searched, but also how the documents were analysed. Just a few sentences are needed to describe the methods.
  • 287: It is unclear to me who is covering the costs of infrastructure here.
  • Section 3.1: Degree 3516 both changed land use and land value. This section would benefit on an elaboration of corresponding changes in land value in these areas alongside a description of what happened to land/property taxation during these years. This section would also benefit from some description of the timeframes that are being discussed. It is unclear to me as a reader when these changes were happening and over what time period.
  • Section 3.2: This section presents data from 2017, providing a descriptive snapshot which is very useful for the analysis. It would benefit greatly from an analysis over time – even just two years such as 2007, 2017 in order to understand change over time. A key question here is are these trends accelerating and if so, uniformly? This would connect well to the following section.